# Hydrogen Selective SiCH Inorganic–Organic Hybrid/γ-Al_2_O_3_ Composite Membranes

**DOI:** 10.3390/membranes10100258

**Published:** 2020-09-25

**Authors:** Miwako Kubo, Ryota Mano, Misako Kojima, Kenichi Naniwa, Yusuke Daiko, Sawao Honda, Emanuel Ionescu, Samuel Bernard, Ralf Riedel, Yuji Iwamoto

**Affiliations:** 1Department of Life Science and Applied Chemistry, Graduate School of Engineering, Nagoya Institute of Technology, Gokiso-cho, Showa-ku, Nagoya 466-8555, Japan; m.kubo.579@stn.nitech.ac.jp (M.K.); r.mano.012@nitech.jp (R.M.); m.kojima.811@stn.nitech.ac.jp (M.K.); k.naniwa.270@stn.nitech.ac.jp (K.N.); daiko.yusuke@nitech.ac.jp (Y.D.); honda@nitech.ac.jp (S.H.); 2Institut für Materialwissenschaft, Technische Universität Darmstadt, Otto-Berndt-Str. 3, 64287 Darmstadt, Germany; ionescu@materials.tu-darmstadt.de (E.I.); ralf.riedel@tu-darmstadt.de (R.R.); 3CNRS, IRCER, UMR 7315, University of Limoges, F-87000 Limoges, France; samuel.bernard@unilim.fr

**Keywords:** allyl-hydrido-polycarbosilane (AHPCS), organic–inorganic hybrid, hydrophobicity, membrane, hydrogen separation, hydrogen affinity, polymer-derived ceramics (PDCs)

## Abstract

Solar hydrogen production via the photoelectrochemical water-splitting reaction is attractive as one of the environmental-friendly approaches for producing H_2_. Since the reaction simultaneously generates H_2_ and O_2_, this method requires immediate H_2_ recovery from the syngas including O_2_ under high-humidity conditions around 50 °C. In this study, a supported mesoporous γ-Al_2_O_3_ membrane was modified with allyl-hydrido-polycarbosilane as a preceramic polymer and subsequently heat-treated in Ar to deliver a ternary SiCH organic–inorganic hybrid/γ-Al_2_O_3_ composite membrane. Relations between the polymer/hybrid conversion temperature, hydrophobicity, and H_2_ affinity of the polymer-derived SiCH hybrids were studied to functionalize the composite membranes as H_2_-selective under saturated water vapor partial pressure at 50 °C. As a result, the composite membranes synthesized at temperatures as low as 300–500 °C showed a H_2_ permeance of 1.0–4.3 × 10^−7^ mol m^−2^ s^−1^ Pa^−1^ with a H_2_/N_2_ selectivity of 6.0–11.3 under a mixed H_2_-N_2_ (2:1) feed gas flow. Further modification by the 120 °C-melt impregnation of low molecular weight polycarbosilane successfully improved the H_2_-permselectivity of the 500 °C-synthesized composite membrane by maintaining the H_2_ permeance combined with improved H_2_/N_2_ selectivity as 3.5 × 10^−7^ mol m^−2^ s^−1^ Pa^−1^ with 36. These results revealed a great potential of the polymer-derived SiCH hybrids as novel hydrophobic membranes for purification of solar hydrogen.

## 1. Introduction

Hydrogen (H_2_) is an attractive energy carrier because of its high energy yield of 120 J g^−1^. This value is about 2.8 times higher than hydrocarbon fuels [1]. Moreover, the combustion product is water and thus completely clean in a carbon dioxide (CO_2_)-neutral manner.

In addition to the current hydrogen production by the steam reforming of naphtha and methane, hydrogen can be produced using several resources including hydropower, nuclear energy and renewable energy sources like biomass, wind, geothermal and solar. Among them, hydrogen production via photoelectrochemical (PEC) water-splitting has received increased attention as an environmental-friendly and low-cost solar-to-hydrogen pathway because of its potential for high conversion efficiency at low operating temperatures using cost-effective semiconductor-based photoreaction catalysts [2,3,4]. The solar hydrogen production systems are expected to be used in terms of global-warming prevention and a stable supply of energy. Therefore, in Japan, the research and development of these systems have been promoted by the Ministry of Economy, Trade and Industry (METI). For example, as an ongoing New Energy and Industrial Technology Develop Organization (NEDO) R&D project, the “Artificial Photosynthesis” Project has been conducted by the Japan Technological Research Association of Artificial Photosynthetic Chemical Process (ARPChem) [5,6]. This project is composed of three research teams: the Solar Hydrogen Team for water photo-splitting catalysts, the Hydrogen Separation Team for gas separation membranes and the Synthetic Catalyst Team for CO_2_ hydrogenation catalysts. All teams are concentrating in materials research and development. These teams also have their missions for future planning and designing industrial photocatalytic plants, gas separation systems with safety measures against explosion, and catalytic synthesis plants, respectively [5,6].

The PEC reaction simultaneously generates H_2_ and oxygen (O_2_) by an oxidation–reduction reaction of water under sunlight in the presence of semiconducting catalysts
(1)$$2H_{2}O\;\to \;2H_{2}\;+{\;O}_{2}$$

Since H_2_ and O_2_ react in a large range of H_2_ concentration of 4–95% [7], this hydrogen production system requires an efficient separation technology for purifying H_2_ from the syngas containing O_2_.

There are several candidate technologies for H_2_ purification such as cryogenic distillation, PSA (pressure swing adsorption) and membrane separation. Among them, membrane separation shows some advantages including energy efficiency and simple separation schemes suitable for establish safe operation process for purification of solar H_2_.

However, under the given operating conditions [5,6,8], it is difficult to apply conventional H_2_-selective membranes, which require higher operation temperatures, approximately above 90, 180 and 350 °C for polymer [9], silica [10,11,12] and metal membranes [13,14,15], respectively. There are also technical issues such as water-induced swelling for polymer membranes, lower H_2_ permeance for supported liquid membranes [16,17,18] and gas permeability degradation for intrinsically hydrophilic silica-based membranes [8,19,20].

Recently, increasing attention has been directed to organic–inorganic hybrid materials as promising functional materials in various fields such as optics, electronics and energy. Synergistic properties of hybrid materials can be achieved by harmonizing advantageous properties of an organic component, such as solubility, plasticity and hydrophobicity, with those of an inorganic component such as high strength and thermal and/or chemical stability [21,22]. The polymer-derived ceramics (PDCs) route [23,24] appears to be an efficient approach for the synthesis of novel organic–inorganic hybrids categorized as Class II hybrids according to the classification given by Sanchez [25], where the organic and inorganic components are linked together by strong chemical bonds, by polymerization and thermal or chemical cross-linking of silicon-based polymers. Moreover, this process can provide a means to vary the specific properties of the silicon-based polymers such as solubility and viscosity, which provides the versatility in shaping capabilities including the formation of surface coatings and membranes similar to those successfully achieved with hydrocarbon-based polymers. In the topic of PDCs, polycarbosilanes (PCSs) are well known as precursors for silicon carbide (SiC)-based ceramics [26,27]. Their synthesis by the Kumada rearrangement of polydimethylsilane (PDS) has been reported by Yajima et al. [26]. Since polymer-derived amorphous silicon (oxy)carbide (SiC and SiCO) show excellent thermal and chemical stabilities [28,29], PDS [30], PCSs [31,32,33,34,35,36,37] and PCS derivatives such as allyl-hydro-polycarbosilane (AHPCS) [38,39,40,41,42] have been applied as precursors of microporous amorphous SiOC [30,31,32,33,34] and SiC [29,35,36,37,38,39,40,41,42] membranes to investigate their gas permeation properties mainly at high temperatures, *T* ≥ 200 °C. PCSs are also appropriate as functional Class II hybrids: The silicon-carbon (Si-C) backbone endows several attractive properties of PCSs such as high flexibility and excellent thermal, chemical and electrical stabilities [43,44,45]. The hydrocarbon groups attached to the Si-C backbone provide room temperature stability in air and PCSs can be applied as an active binder for the formation of powder compact to fabricate polycrystalline SiC ceramics [46,47,48,49]. Moreover, in our previous study [50], PCSs were found to show an excellent hydrophobic property: gas permeations under highly humid condition at 50 °C of a hydrophilic mesoporous γ-Al_2_O_3_ membrane were successfully stabilized by modification with SiCH organic–inorganic hybrid via melt impregnation of PCS at 120 °C [50].

In this study, AHPCS was converted to highly cross-linked ternary SiCH organic–inorganic hybrids with enhanced thermal stability and hydrophobicity by heat treatment at temperatures as low as 300–500 °C in argon (Ar). Highly hydrophobic property of the AHPCS-derived SiCH hybrids was characterized by the water vapor adsorption–desorption isotherm measurement at 25 °C. Then, the SiCH organic–inorganic hybrid served to modify a mesoporous γ-Al_2_O_3_ membrane via dip-coating with AHPCS followed by the heat treatment of the composite at 300–500 °C in Ar.

Single gas permeances of helium (He), hydrogen (H_2_) and nitrogen (N_2_) under dry condition at 25–80 °C were measured to characterize the intrinsic low-temperature gas permeation properties of the SiCH organic–inorganic hybrid/γ-Al_2_O_3_ composite membrane. Then, as the primary accelerated degradation test for the suggested solar H_2_ purification condition of 10% humidity at 50 °C [5,6], the gas permeance measurements were performed under the saturated water vapor partial pressure at 50 °C. In this measurement, a H_2_-N_2_ mixed feed gas in the molar ratio 2:1 was used as a simulated syngas produced by the PEC reaction (Equation (1)) in which O_2_ (0.346 nm) [51] was replaced by inert N_2_ having a similar kinetic diameter (0.364 nm) [51] to avoid explosion accidents. Moreover, for some selected membrane samples and the supported mesoporous γ-Al_2_O_3_ membrane itself, cyclic gas permeation measurements were continuously performed under water vapor partial pressures ranging from 0 to 1.0 at 50 °C. Effect of the composite membrane synthesis temperature on the hydrophobicity and gas permeation properties under dry and wet conditions were discussed aiming to develop novel hydrophobic membranes for the purification of solar hydrogen at low temperatures around 50 °C.

## 2. Experimental Procedures

### 2.1. Preparation of the Supported Mesoporous γ-Al_2_O_3_ Membrane

Commercially available macroporous α-Al_2_O_3_ tubular support (6 mm outer diameter, 4 mm inner diameter and 60 mm length, Noritake Co., Ltd., Nagoya, Japan) was used. The α-Al_2_O_3_ support was composed of a fine outer surface layer (mean pore diameter, *d*_p_ = 150 nm) and a core (*d*_p_ = 700 nm). Total porosity of the tubular support was 40%. A mesoporous γ-Al_2_O_3_ membrane was fabricated by dip coating of a boehmite (γ-AlOOH) sol on the α-Al_2_O_3_ tubular support followed by heat treatment in air at 600 °C for 3 h according to a published procedure [50].

### 2.2. Modification of a Supported γ-Al_2_O_3_ Membrane with AHPCS-Derived Organic–Inorganic SiCH Hybrid

Commercially available AHPCS (SMP-10 was provided by Starfire Systems, Inc., Glenville, NY, USA. Anal. Found (wt %): Si, 54.8; C, 35.0; H, 8.3; O, 1.9. FT-IR (ATR/cm^−1^): ν (C−H) = 3076 (s), 3003 (s), 2958 (s), 2895 (s), 2850 (m), ν(Si−H) = 2123 (vs), δ(allyl) = 1629 (m), δ(CH_2_) = 1395 (m), δ(Si-CH_3_) = 1251 (s), δ(Si-CH_2_-Si) = 1049 (s), δ(Si-H) = 941.1 (s), δ(Si-C) = 854.3 (s), δ(SiCH_3_) = 770 (s); ^1^H NMR (300 MHz, CDCl_3_, δ/ppm): 0.05 (br, -Si-C**H**_3_), 0.4 (br, -Si-C**H**_2_-), 1.56–1.9 (br, -Si-C**H**_2_-CH=CH_2_), 3.4–3.84 (br, -Si**H**_3_C), 3.8–4.1 (br, -Si**H**_2_CH_2_), 4.85–5.05 (br, -Si-CH_2_-CH=C**H**_2_), 5.71–5.93 (br, Si-CH_2_-C**H**=CH_2_). As a precursor solution for modifying the supported γ-Al_2_O_3_ membrane, 1 wt % dry xylene (super dehydrated grade, 99.5% purity, Wako Pure Chemical Co., Ltd., Osaka, Japan) solution of as-received AHPCS was prepared under the inert atmosphere of Ar. By following the procedures described in our previous report [50], the AHPCS solution was dip-coated on the supported γ-Al_2_O_3_ membrane, then dried and heat-treated under flowing Ar at 300, 400 and 500 °C for 1 h at a heating/cooling rate of 100 °C h^−1^ to afford SiCH organic–inorganic hybrid-modified γ-Al_2_O_3_ membrane placed on the α-Al_2_O_3_ tubular support (SiCH hybrid/γ-Al_2_O_3_ composite membrane).

The 500 °C-heat treated membrane was further modified by the melt impregnation of a commercially available PCS at 120 °C under Ar atmosphere (Type L, Nippon Carbon Co., Ltd., Tokyo, Japan) according to the published procedure [50].

### 2.3. Characterizations

The molecular weight distribution curve of as-received AHPCS was measured at 40 °C by using gel permeation chromatography (GPC, Model ShodexGPC-104 equipped with two tandem columns (Model Shodex LF-404, Showa Denko K.K., Tokyo, Japan) and a refractive index detector (Model Shodex RI-74S, Showa Denko K.K., Tokyo, Japan). The columns were calibrated against polystyrene standards. Tetrahydrofuran (THF, 99.5% purity, Wako Pure Chemical Co., Ltd., Osaka, Japan) was used as the eluent and a flow rate was adjusted to 1.0 mL min^−1^.

The thermal decomposition and cross-linking behaviors of as-received AHPCS up to 1000 °C was studied by thermogravimetry combined with mass spectrometry (TG-MS) analyses (Model STA7200, Hitachi High Technologies Ltd., Tokyo, Japan/Model JMS-Q1500 GC, JEOL, Tokyo, Japan). The measurements were performed under He atmosphere with a heating rate of 10 °C min^−1^.

Powder samples of the heat-treated AHPCS were prepared under the same manner for the heat treatment of the supported mesoporous γ-Al_2_O_3_ membrane after dip-coating of the AHPCS solution.

Fourier transform (FT)-IR spectrum was recorded on the as-received AHPCS and AHPCS-derived powder samples by the potassium bromide (KBr) disk method (Model FT/IR-4200IF, JASCO Corp., Tokyo, Japan). Note that FT-IR spectrum was also recorded on the powder sample of 700 °C-heat treated AHPCS. Raman spectrum was recorded on as-received AHPCS and heat-treated AHPCS (Renishaw, inVia Reflex, New Mills, England).

Hydrophobicity of the heat-treated AHPCS powder samples was characterized by measuring the water vapor adsorption–desorption isotherms at 25 °C (Model BELSORP-aqua 3, MicrotracBEL Corp., Osaka, Japan). The powder samples were pretreated at 120 °C for 6 h under vacuum.

The cross-sectional structure and top surface of the SiCH hybrid/γ-Al_2_O_3_ composite membrane were observed by a scanning electron microscope (SEM, Model JSM-6360LV, JEOL Ltd., Tokyo, Japan). The distribution of the SiCH hybrid within the composite membrane was examined by the energy dispersive X-ray spectroscopic (EDS) analysis (Model JSM-6010LA mounted on SEM, JEOL Ltd., Tokyo, Japan).

Single gas permeances through the membrane samples under dry conditions were measured for He, H_2_ and N_2_ by the volumetric method at constant pressure. The setup of the equipment and procedure for the measurements was shown in our previous report [50]. The gas permeances at 25, 50 and 80 ˚C were measured in the size order of a kinetic diameter for He (0.26 nm), H_2_ (0.289 nm) and N_2_ (0.364 nm) [51]. The single gas permeance (*Q_i_*) was evaluated by using Equation (2),
(2)$$Q_{i}=\frac{V}{A\left(p_{H}-p_{L}\right)}$$
where *V* (mol s^−1^) is the permeate molar flow rate, *A* (m^2^) is the membrane area and *p_H_* (Pa) and *p_L_* (Pa) are pressures of the gas feed side and the gas permeate side, respectively. In this study, the (*p_H_ − p_L_*) in Equation (1) was fixed as 100 kPa.

The permselectivity (α) was evaluated by calculating the single gas permeance ratio of two different kinds of gases.

H_2_ and N_2_ gas permeances under dry and wet condition of the saturated water vapor partial pressure (p/p_0_ (H_2_O) = 1.0) were measured at 50 °C using a mixed feed gas with a 2:1 molar ratio of H_2_ and N_2_ as a simulated syngas produced by the PEC reaction. For some selected membranes and the supported mesoporous γ-Al_2_O_3_ membrane itself, cyclic gas permeation measurements were performed under various p/p_0_ (H_2_O) ranging from 0 to 1.0 at 50 °C.

The permeate gas composition was analyzed using a gas chromatograph (GC, Model CP-4900 Micro-GC, Varian medical systems Inc., Palo Alto, CA, USA) and Ar sweep gas (50 mL min^−1^). Each gas permeance of H_2_ and N_2_ was calculated using the analyzed composition and the measured mixed gas permeate molar flow rate. The water vapor permeance was evaluated using a gas chromatograph for the polar gas analysis (GC323, GL Sciences Inc., Tokyo, Japan) according to the published procedure [50].

The membrane performance under the H_2_-N_2_ (2:1) mixed feed gas flow at p/p_0_ (H_2_O) = 1.0 was assessed in terms of the separation factor (*SF*):(3)$$SF=\frac{Y_{A}/Y_{B}}{X_{A}/X_{B}}$$
where *Y* and *X* are the mass fractions of permeate and feed, respectively and subscripts *A* and *B* denote H_2_ and N_2_, respectively.

To examine the hydrogen affinity of the AHPCS-derived SiCH organic–inorganic hybrids, the amount of H_2_ adsorption onto a thin film of the AHPCS-derived hybrid was measured using a quartz-crystal microbalance (QCM), which is one of the useful methods for detecting in-situ minute mass changes [52]. Commercially available quartz crystal unit (SEN-9E-H-10, Tamadevice Co., Ltd., Kawasaki, Japan) connected with a crystal oscillator circuit (Tamadevice Co. Ltd., Kawasaki, Japan), power supply (GPS-S, GW Instek, Texio Technology Co., Yokohama, Japan) and frequency counter (SC-7205A, UNIVERSALCOUNTER, Iwatsu Electric Co., Ltd., Tokyo, Japan) were used, and the measurement system shown in Figure 1 was set-up at our laboratory.

The sample film was formed on the quartz crystal unit surface using the dry xylene solution of as-received AHPCS under the same manner as mentioned for the 300 °C-synthesized membrane. The sample film on the quartz crystal unit was placed in a four-necked round-bottom glass flask. Then, the temperature inside the flask was precisely adjusted at 30 °C (±0.1 °C) by a styrofoam-covered mantle heater operated by the PID control. The sample film was exposed to He atmosphere maintained by a continuous He gas flow at 30 mL min^−1^. After 15 h, the frequency shift was measured for 90 ks to confirm that the He adsorption reached the equilibrium. Then, the weight change measurement under a H_2_ flow (30 mL min^−1^) was started by switching the feed gas from He to H_2,_ and the weight change was monitored for an additional 90 ks. The measurement system was controlled by a standard PC with a software (Labview, National Instruments Corp., Austin, TX, USA) for recording the transition of the frequencies corresponding to the mass change.

Conversion from the measured frequency shift Δ*f* (Hz) to the mass change Δ*m* (g) was calculated using the Sauerbrey equation [53],
(4)$$\Delta f=-\frac{2f_{0}^{2}}{\sqrt{\mu_{q\;}\rho_{q}}}\;\frac{\Delta m}{A_{e}}$$
where, *f*_0_ is the frequency of the quartz crystal prior to a mass change (9.0 × 10^6^ (Hz)), *μ*_q_ is the shear modulus of quartz (2.947 × 10^13^ (g m^−1^ s^−2^)), *ρ*_q_ is the density of quartz (2.648 (g cm^−3^)) and *A_e_* is the electrode area (3.93 × 10^−5^ (m^2^)).

## 3. Results and Discussion

### 3.1. Heat Treatment Temperatures Selected for Highly Cross-Linked SiCH Organic–Inorganic Hybrid Synthesis

Chemical structure and molecular distribution of the AHPCS are shown in Appendix A, respectively, in the Appendix A. In this study, the SiCH hybrid polymer of AHPCS was converted into the highly cross-linked SiCH organic–inorganic hybrid as a component of the hydrogen separation membrane. The temperatures for thermal conversion of AHPCS to highly cross-linked SiCH hybrid in this study was selected as 300, 400 and 500 °C based on the results obtained by the TG-MS analyses (Appendix A) as well as FT-IR and Raman spectroscopic analyses for the heat-treated AHPCSs (Appendix A). The thermal behavior of AHPCS has been already studied by several research groups [42,54,55,56,57,58], and the results obtained in this study were well consistent with those previously reported: As shown in Appendix A, as-received AHPCS had a considerable amount of low molecular weight fraction below 1000. TG-MS analyses revealed the thermal decomposition of the low molecular weight fraction proceeded during the first weight loss at 100–300 °C and a second one from 350 to 500 °C by detecting gaseous species assigned to the fragments of carbosilane species (Appendix A). On the other hand, thermal cross-linking was observed up to 300 °C for the formation of ≡Si-CH_2_-CH_2_-CH_2_-Si≡ and/or ≡Si-CH(CH_3_)-CH_2_-Si≡ via hydrosilylation between ≡Si-H and ≡Si-CH_2_-CH=CH_2_ groups in AHPCS, which was identified by the disappearance of the FT-IR absorption band at 1629 cm^−1^ attributed to the C=C bond of the allyl group [55,56] associated with the decrease in the relative FT-IR band intensities assigned to ν(Si-H) at 2123 cm^−1^ and δ(Si-H) at 947 cm^−1^ [55,56] (Appendix A). At 400–700 °C, formation of ≡Si-Si≡ between Si-H and Si-CH_3_ groups [42] was suggested by detecting gaseous species at the m/z ratio of 15 assigned to methane (CH_4_; Appendix A). Since the thermal decomposition and cross-linking contentiously proceeded at 300–500 °C, the quantity of organic groups and microporosity of the SiCH hybrid differed depending on the specific heat treatment temperature in this temperature range. On the other hand, the FT-IR spectrum for the 700 °C-heat treated AHPCS revealed that polymer/inorganic silicon carbide conversion almost completed (Appendix A). It should be noted that the samples heat-treated at 300–500 °C were identified as the ternary SiCH Class II hybrid without graphite-like carbon, since the Raman spectra of these samples exhibited several peaks due to the organic groups without those attributed to graphite-like carbon typically detected at 1347.5 and 1596.5 cm^−1^ assigned as D-band (for disordered graphite) and G-band (for the sp2 graphite network), respectively [59,60] (Appendix A). Based on these results, the heat treatment of AHPCS was achieved at 300, 400 and 500 °C for the synthesis of powder and membrane samples.

### 3.2. Hydrophobicity

The water vapor adsorption–desorption isotherms at 25 °C measured for the powder samples are shown in Figure 2. The AHPCS heat-treated at 300 and 400 °C generated a type III isotherm [61,62], which showed a weak interaction to water molecule. The maximum amount of the water adsorption ((Va(H_2_O)) of these samples was below 2 cm^3^(STP) g^−1^. The 500 °C-heat treated AHPCS also showed relatively high hydrophobicity, and the Va(H_2_O) remained at 6.3 cm^3^ (STP) g^−1^.

The Va(H_2_O) values evaluated for the AHPCS-derived SiCH were much lower than that of highly hydrophilic mesoporous γ-Al_2_O_3_ (297 cm^3^ (STP) g^−1^) characterized in our previous study [50].

On the other hand, the 700 °C-heat treated AHPCS presented a type V [61,62]-like isotherm with a large open loop of hysteresis at p/p_0_ (H_2_O) above 0.3, and the Va(H_2_O) reached 42.5 cm^3^ (STP) g^−1^. As shown in Appendix A, the thermal decomposition of the organic groups in AHPCS was almost completed at 700 °C, which led to the analyzed hydrophilic character.

### 3.3. Properties of SiCH Hybrid/γ-Al_2_O_3_ Composite Membrane

(1)Structure of the composite membrane

As a typical observation, Figure 3a presents a cross-sectional SEM image of a supported γ-Al_2_O_3_ membrane after dip coating of the AHPCS xylene solution and subsequent heat treatment at 400 °C under flowing Ar. There was no additional layer on the γ-Al_2_O_3_ membrane surface. Then, an EDS analysis was performed on the modified membrane. As shown in Figure 3b, the line scan of the EDS mapping for Si derived form AHPCS was detected within the γ-Al_2_O_3_ membrane having approximately 2.5 μm thickness. Accordingly, the resulting composite membranes in this study were composed of γ-Al_2_O_3_ with mesopore channels infiltrated by the AHPCS-derived SiCH hybrid.

(2)Gas permeation behaviors under dry condition

Arrhenius plots of He, H_2_ and N_2_ permeances evaluated for the SiCH hybrid/γ-Al_2_O_3_ composite membranes synthesized at 300, 400 and 500 °C are shown in Figure 4a–c, respectively. Despite the low permeation temperature of 25 °C, the composite membranes exhibited a relatively high H_2_ permeance of 1 × 10^−7^ to 4 × 10^−6^ mol m^−2^ s^−1^ Pa^−1^ with a H_2_/N_2_ permselectivity (α(H_2_/N_2_)) of 9.2–17, apparently higher than that of the theoretical one (3.73) based on the Knudsen’s diffusion.

The gas permeation behavior through each composite membrane was similar, and all the gas permeances increased linearly with the permeation temperature and thus followed the Arrhenius law. Since the gas permeations through the supported mesoporous γ-Al_2_O_3_ membrane exhibited a typical Knudsen’s diffusion characteristics in our previous study [50], the gas permeation behaviors observed for the present composite membranes suggested that all the gases permeated through the microporous SiCH hybrid, which filled in the mesopore channels of the γ-Al_2_O_3_, and the dominant mechanism for the gas permeations was activated diffusion. However, at all permeation temperatures from 25 to 80 °C, the composite membranes exhibited a unique α(H_2_/He) of 1.44–1.95, which was higher than the theoretical one (1.41) based on the Knudsen’s diffusion.

To highlight this unique gas permeation behavior, the gas permeances measured at 50 °C were characterized and are shown in Figure 5 by plotting the kinetic diameter dependence of the normalized gas (*i*) permeance relative to the He permeance (*Q*_i_/*Q*_He_), and compared with the ideal value (*Q_K_*_,i_/*Q_K_*_,He_) by the Knudsen model [63,64],
(5)$$\frac{Q_{K,i}}{Q_{K,\mathrm{He}}}=\frac{\sqrt{M_{\mathrm{He}}}}{\sqrt{M_{i}}}\;\;\;\;\;\;$$
where *M*_i_ and *M*_He_ are molecular weights of gas-*i* and He, respectively.

The apparent activation energies for He and H_2_ permeations were evaluated by the Arrhenius plot (Figure 4) and are listed in Table 1. The apparent activation energy (*E_a_*) for He permeation was 20.3–1.4 kJ mol^−1^, while that for H_2_ permeation was 17.0–0.5 kJ mol^−1^.

The *E_a_* values for both He and H_2_ permeations decreased with the composite membrane synthesis temperature. This tendency was well consistent with the result of TG-MS analyses (Appendix A) and the gas permeation behaviors (Figure 4): weight loss due to the volatilization of the low molecular weight fraction of as-received AHPCS contentiously proceeded at 300–500 °C (Appendix A), and resulting yields of the SiCH hybrid at 300, 400 and 500 °C were measured to be 87%, 83% and 76%, respectively (Appendix A), while all the gas permeances through the composite membrane increased with the synthesis temperature (Figure 4). Thus, the observed *E_a_* values for He and H_2_ permeations could depend on the density of the AHPCS-derived SiCH organic–inorganic hybrid network. However, a unique point was that, regardless of the synthesis temperature, the *E_a_* for H_2_ permeation was found to be smaller than that for He permeation, which was a clear contrast to those evaluated for the H_2_-selecitve microporous amorphous silica membranes [10,11,65,66,67]. Then, an attempt was made for study on the hydrogen affinity of the AHPCS-derived SiCH organic–inorganic hybrid: The amount of H_2_ adsorption on the AHPCS-derived SiCH film was measured by using a quartz-crystal microbalance (QCM). The SiCH sample film was formed on the quartz crystal unit surface by following the procedure for the composite membrane synthesis at 300 °C (effective film area corresponded to *A_e_* in Equation (2), 3.93 × 10^−5^ m^2^). The sample film was exposed to He atmosphere under the strictly regulated isothermal condition at 30 °C (±0.1 °C) for more than 90 ks. After the He adsorption reached equilibrium, weight gain was monitored by exposing the sample film to H_2_ atmosphere under the same isothermal condition. As shown in Figure 6, the weight gain increased with H_2_ exposure time and reached 7.14 ng (1.82 × 10^−4^ g m^−2^) after the additional 88,130 s. Since the molecular weight of H_2_ (2.016) was approximately one-half of He (4.002), the weight gain measured under the H_2_ atmosphere revealed preferential adsorption of H_2_ relative to He, i.e., existence of H_2_ affinity of the AHPCS-derived SiCH organic–inorganic hybrid, which might contribute to the experimentally observed unique α(H_2_/He) > 1.41 and high H_2_ permeances (10^−7^–10^−6^ mol m^−2^ s^−1^ Pa^−1^ order at 25–80 °C).

(3)Gas permeation behaviors under the wet condition

For study on the potential application to the solar hydrogen production system, the gas permeation measurement was performed on the composite membranes by using a mixed H_2_-N_2_ feed gas in the molar ratio 2:1 under dry and saturated water vapor partial pressure (p/p_0_ (H_2_O) = 1) at 50 °C. The results are summarized and shown in Figure 7. The composite membranes were found to be water vapor permeable and the permeance was measured to be 2.7–3.8 × 10^−7^ mol m^−2^ s^−1^ Pa^−1^.

The α(H_2_/N_2_) of the composite membranes was 6.0–11.3, and regardless of the synthesis temperature, there was no significant degradation under the highly humid condition at 50 °C. The 300 °C-synthesized composite membrane kept relatively high gas permeances. The retentions evaluated for the H_2_ and N_2_ permeances under the p/p_0_ (H_2_O) = 1 at 50 °C were 69 and 67%, respectively. However, with increasing synthesis temperature, the permeance retention of both H_2_ and N_2_ decreased, in other words, the hydrophobicity in terms of stable gas permeation property degraded with the synthesis temperature. This degradation tendency is consistent with the temperature dependence of the SiCH hybrid polymer/highly cross-linked SiCH hybrid conversion yield as described above, and thus is related to the quantity of organic groups remained in the SiCH hybrid and the microporosity or volume of the SiCH hybrid, which infiltrated the mesopore channels of the composite membrane.

Then, the 500 °C-synthesized composite membrane was further modified with polycarbosilane (PCS) by the 120 °C-melt impregnation established by our previous study [50]. As shown in Figure 7, under the dry condition at 50 °C, the PCS-modified composite membrane showed a relatively high H_2_ permeance of 8.4 × 10^−7^ mol m^−2^ s^−1^ Pa^−1^ with a significantly improved α(H_2_/N_2_) of 29.6. Moreover, under the p/p_0_ (H_2_O) = 1 at 50 °C, the PCS-modification successfully improved the membrane performance: H_2_ permeance and α(H_2_/N_2_) were measured to be 3.5 × 10^−7^ mol m^−2^ s^−1^ Pa^−1^ and 36, respectively.

Gas permeation properties under the wet condition at 50 °C of the composite membranes were also assessed by cyclic gas permeance measurements under p/p_0_(H_2_O) ranging from 0.1 to 1.0 and compared with those of the supported mesoporous γ-Al_2_O_3_ membrane itself. As shown in Figure 8a, H_2_ and N_2_ permeances through the supported mesoporous γ-Al_2_O_3_ membrane drastically decreased above p/p_0_(H_2_O) = 0.74. This degradation is due to the highly hydrophilic property of γ-Al_2_O_3_, which leads to the blockage of the gas permeable mesopore channels by adsorption and subsequent condensation of water molecules as the permeate [50]. On the other hand, the 300 °C-synthesized composite membrane exhibited stable gas permeations at all the p/p_0_(H_2_O) up to 1.0 (Figure 8b). A slight decrease in the gas permeances at p/p_0_(H_2_O) > 0.74 was due to the pressure-drop caused by the water vapor condensation within the mesopore channels of hydrophilic γ-Al_2_O_3_ partly remained without surface modification with the AHPCS-derived hydrophobic hybrid. The 500 °C-synthesized composite membrane also showed the decreasing tendency in gas permeances, however further modification with PCS successfully improved the membrane performance at the final gas permeation measurement under the saturated humidity at 50 °C: H_2_ permeance remained at 10^−7^ mol m^−2^ s^−1^ Pa^−1^ order with α(H_2_/N_2_) > 30 under (Figure 8c), and the resulting separation factor (SF) evaluated based on the gas permeation data shown in Table 2 was found as 26.

Moreover, the polymer-derived SiCH organic–inorganic hybrid investigated in this study showed sufficient stability under the present high humidity conditions at 50 °C: Figure 9 presents the top surface view of the 500 °C-synthesized composite membrane modified with PCS before and after the cyclic gas permeation measurements under p/p_0_(H_2_O) up to 1.0 at 50 °C. Compared with the surface of the as-synthesized mesoporous γ-Al_2_O_3_ membrane over an α-Al_2_O_3_ porous support (Figure 9a), the composite membrane exhibited a smooth surface (Figure 9b) and kept the surface without structural degradation after the cyclic gas permeation measurements (Figure 9c). These results revealed that, in addition to hydrogen permselectivity, the modification with the polymer-derived SiCH organic–inorganic hybrid investigated in this study greatly improved the hydrophobicity in terms of stable gas permeations under the saturated water vapor partial pressure at 50 °C.

The H_2_-permselectivities of the composite membrane measured in this study were briefly compared with those recently reported for other membranes composed of various materials systems, and their H_2_ permeation data with α(H_2_/X) (X = N_2_ (0.364 nm) [51] or O_2_ (0.346 nm) [51]) measured under the dry condition at *T* ≤ 50 °C are listed in Table 3 [18,68,69,70,71,72,73,74,75,76,77,78]. Among them, novel ultrathin (9 nm thickness) graphene oxide membrane formed on an anodic oxidized alumina support (#09) exhibited a H_2_ permeance of approximately 1 × 10^−7^ mol m^−2^ s^−1^ Pa^−1^ with α(H_2_/N_2_) of 900 at 20 °C [75,76]. The zeolite imidazolate framework (ZIF) nanosheet membrane (#08) also showed H_2_ permeance of 2.04 × 10^−7^ mol m^−2^ s^−1^ Pa^−1^ with high α(H_2_/N_2_) of 66.6 at 30 °C [73,74]. Among other practical membranes, SiO_2_-based organic–inorganic hybrid membrane (#03) showed the highest H_2_ permeance of 10^−6^ mol m^−2^ s^−1^ Pa^−1^ order, while the α(H_2_/N_2_) remained at 12 [68]. On the other hand, zeolite/CMC (carbon molecular sieve) composite membranes (#04, #05) showed a high α(H_2_/N_2_) of 61-100.2, however the H_2_ permeance at 30 °C was 10^−9^–10^−8^ mol m^−2^ s^−1^ Pa^−1^ order [69,70].

As shown in Figure 4 and Figure 5, in addition to the H_2_/N_2_ selectivity, the present composite membranes exhibited unique H_2_/He selectivity due to the H_2_ affinity of the AHPCS-derived highly cross-linked SiCH organic–inorganic hybrid in the composite membrane (Figure 6). This unique H_2_ preferential permeation property contributed to relatively high H_2_ permeance of 8.4 × 10^−7^ mol m^−2^ s^−1^ Pa^−1^ with α(H_2_/N_2_) of 29.6.

For the application of the purification of solar hydrogen, the long-term stability and robustness of H_2_-selecitve membranes under the humid condition at around 50 °C are practical issues for us to pursue. In this context, there are few reports as mentioned above, and Oyama et al. reported [18] their pioneering study on the stability of PFDA-based liquid membrane (#06 in Table 3) under the humid condition (10 mol%) at 30 °C as a simulated condition for the purification of solar hydrogen, and they confirmed its stability for up to 48 h. The stabilities of the composited membranes characterized by the primary accelerated degradation test (Figure 7, Figure 8 and Figure 9) were compatible with that of the supported liquid membrane (#06 in Table 3) [18]. Under the scheme of the current NEDO R&D “Artificial Photosynthesis” Project, we plan to conduct the long-term stability test for the composite membranes by using a H_2_-O_2_ (2:1) mixed feed gas as a simulated syngas at the project facility with safety measures against explosion.

## 4. Conclusions

In this study, AHPCS was converted to highly cross-linked ternary SiCH organic–inorganic hybrid compounds by heat treatment at 300–500 °C under Ar atmosphere. The water vapor adsorption–desorption isotherm measurement revealed that the AHPCS-derived highly cross-linked SiCH hybrids exhibited excellent hydrophobicity, and the maximum amount of water vapor adsorption at 25 °C was 1.5–6.3 cm^3^ (STP) g^−1^.

Aiming to develop H_2_-selective membranes for the application of the novel solar hydrogen production system, a supported mesoporous γ-Al_2_O_3_ membrane with a thickness of about 2.5 μm was modified with AHPCS xylene solution and subsequently heat-treated at 300–500 °C under Ar atmosphere. SEM observation revealed that the mesoporous channels of the γ-Al_2_O_3_ membrane were coated by the AHPCS-derived SiCH to afford a SiCH hybrid/γ-Al_2_O_3_ composite membrane. Even at a low temperature of 25 °C, the composite membranes exhibited a high H_2_ permeance of 1 × 10^−7^ to 4 × 10^−6^ mol m^−2^ s^−1^ Pa^−1^ and a α(H_2_/N_2_) of 9.2–17 together with a unique α(H_2_/He) of 1.44–1.95.

The apparent activation energy for the H_2_ permeation at 25–80 °C was 17.0–0.5 kJ mol^−1^ and found to be smaller than that for He (20.3–1.4 kJ mol^−1^). Moreover, the measurement of H_2_ adsorption on an AHPCS-derived SiCH film by using a QCM revealed preferential H_2_ adsorption at 30 °C. These results strongly indicate a significant H_2_ affinity of the AHPCS-derived SiCH organic–inorganic hybrid, which contributes to the experimentally observed unique α(H_2_/He) and the high H_2_ permeance at 25–80 °C.

As a simulated wet condition for the purification of solar hydrogen, the gas permeation measurement under saturated water vapor partial pressure at 50 °C was performed on the composite membranes by using a mixed H_2_-N_2_ feed gas in the molar ratio 2:1. The 300–500 °C synthesized composite membranes exhibited a relatively high H_2_ permeance of 1.0–4.3 × 10^−7^ mol m^−2^ s^−1^ Pa^−1^ with a α(H_2_/N_2_) of 6.0–11.3. Further modification by the 120 °C-melt impregnation of PCS successfully improved the H_2_-permselectivity of the 500 °C synthesized composite membrane by maintaining the H_2_ permeance combination with improved α(H_2_/N_2_) as 3.5 × 10^−7^ mol m^−2^ s^−1^ Pa^−1^ and 36. These results clearly revealed a promising potential of polymer-derived SiCH organic–inorganic hybrids to develop advanced H_2_ selective membranes applicable to novel solar hydrogen production systems.

## Figures and Tables

**Figure 1 membranes-10-00258-f001:**
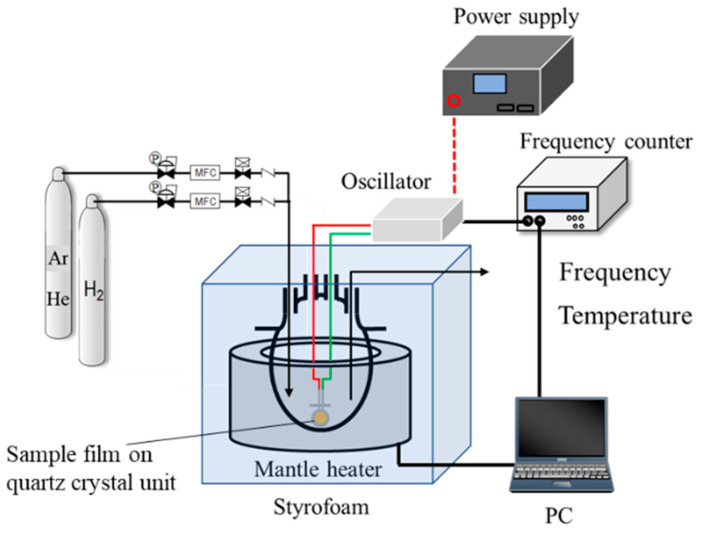
Schematic diagram of the weight change measurement with the quartz crystal microbalance (QCM).

**Figure 2 membranes-10-00258-f002:**
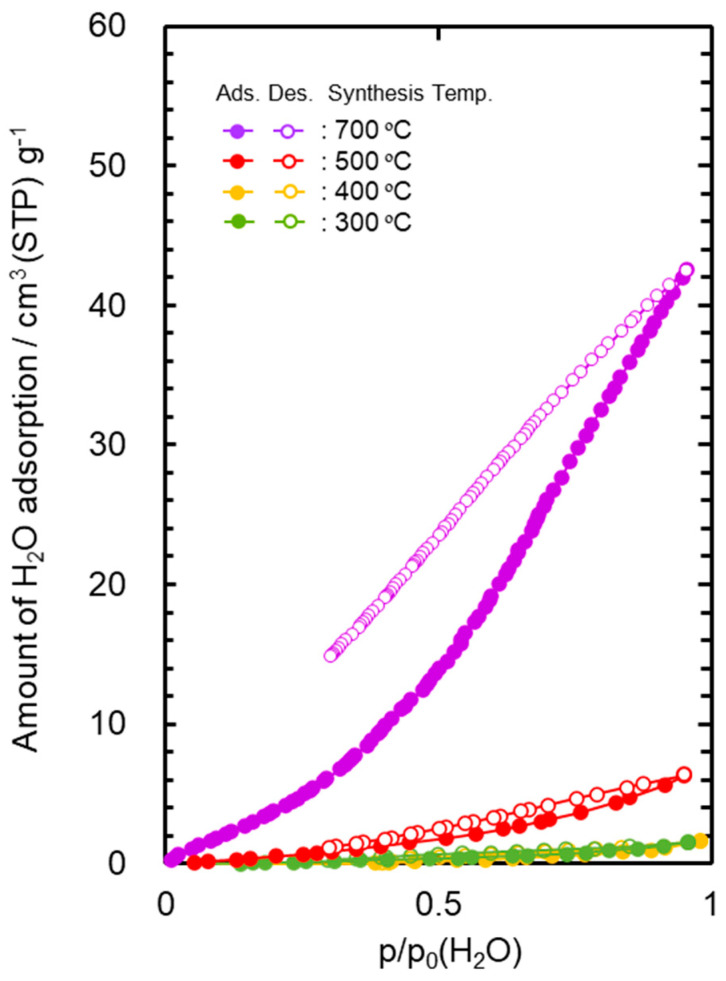
Water vapor adsorption–desorption isotherms at 25 °C for SiCH inorganic–organic hybrid powder samples synthesized by heat treatment of AHPCS at 300–700 °C in Ar.

**Figure 3 membranes-10-00258-f003:**
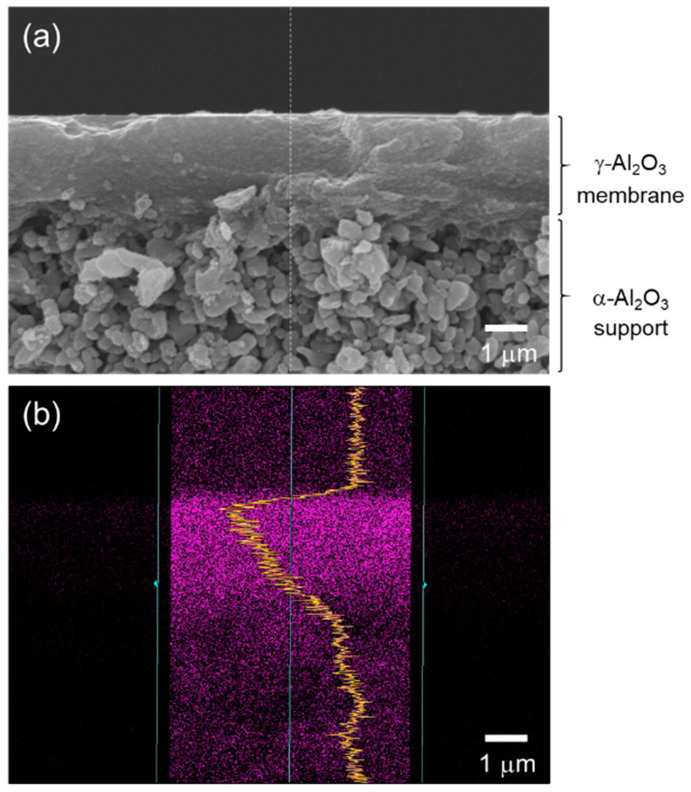
(**a**) A typical cross-sectional SEM image of the γ-Al_2_O_3_ membrane on a microporous α-Al_2_O_3_ support after modification with AHPCS and subsequent heat treatment at 400 °C in Ar. (**b**) Line scan of EDS mapping for Si derived from AHPCS detected within the mesoporous γ-Al_2_O_3_ membrane having a thickness of approximately 2.5 μm.

**Figure 4 membranes-10-00258-f004:**
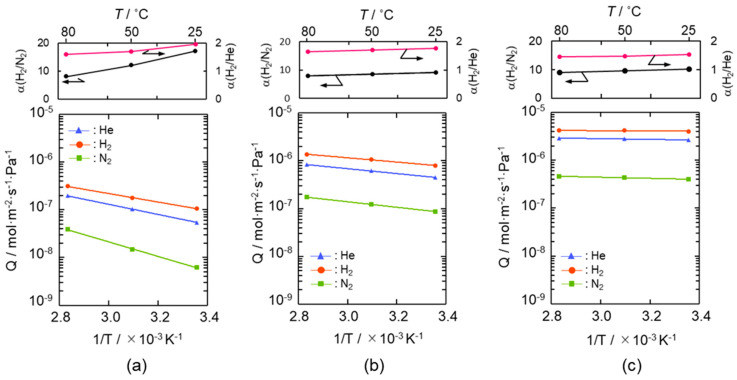
Gas permeation behaviors under dry condition of the supported mesoporous γ-Al_2_O_3_ membrane after modification with AHPCS and subsequent heat treatment under flowing Ar at (**a**) 300 °C, (**b**) 400 °C and (**c**) 500 °C.

**Figure 5 membranes-10-00258-f005:**
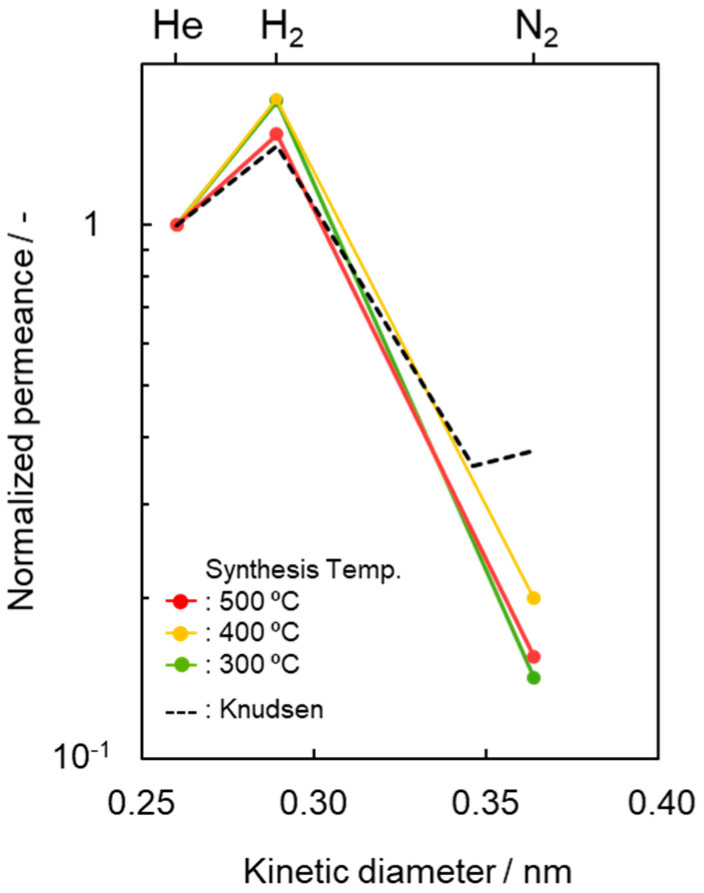
Kinetic diameter dependence of normalized gas permeance based on He permeance (*Q*_i_/*Q*_He_) at 50 °C through the SiCH hybrid/γ-Al_2_O_3_ composite membrane synthesized at different temperatures of 300, 400 and 500 °C. Dotted line indicates predicted values by the Knudsen diffusion model based on He permeance.

**Figure 6 membranes-10-00258-f006:**
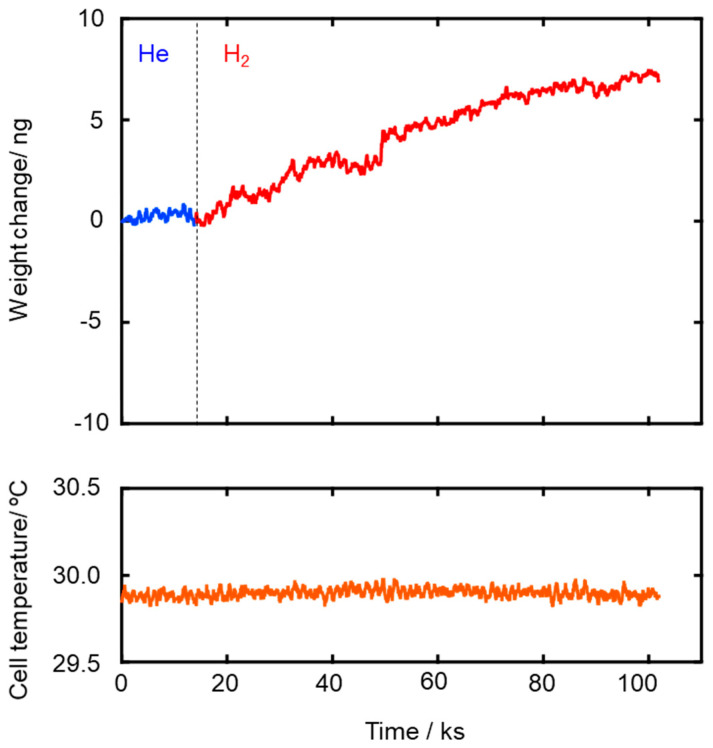
Weight gain measured at 30 °C for H_2_ adsorption to the sample film of the SiCH organic–inorganic hybrid prepared by 300 °C-heat treatment in Ar.

**Figure 7 membranes-10-00258-f007:**
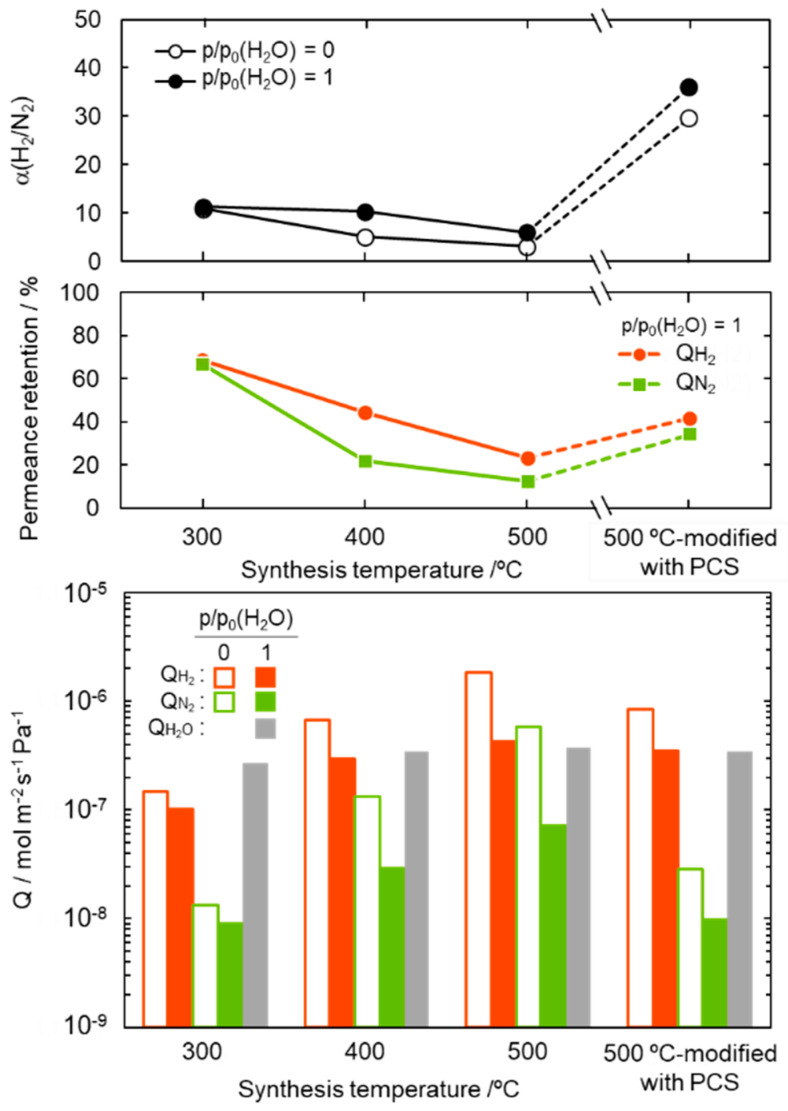
Gas permeation properties under the dry (p/p_0_(H_2_O) = 0) and wet (p/p_0_(H_2_O) = 1) condition at 50 °C, and water vapor permeation properties at p/p_0_(H_2_O) = 1 at 50 °C evaluated for SiCH hybrid/γ-Al_2_O_3_ composite membranes synthesized at different temperatures of 300, 400 and 500 °C, and the 500 °C-synthesized membrane further modified with PCS by the melt impregnation at 120 °C [50].

**Figure 8 membranes-10-00258-f008:**
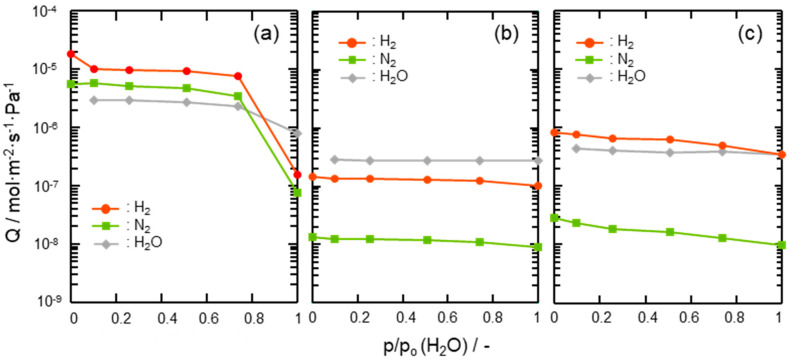
Results of cyclic gas permeation measurements at 50 °C under p/p_0_(H_2_O) = 0–1.0 using the H_2_-N_2_ (2:1) mixed feed gas evaluated for (**a**) a supported mesoporous γ-Al_2_O_3_ membrane, (**b**) a 300 °C-synthesized SiCH hybrid/γ-Al_2_O_3_ composite membrane and (**c**) a 500 °C-synthesized composite membrane further modified with PCS.

**Figure 9 membranes-10-00258-f009:**
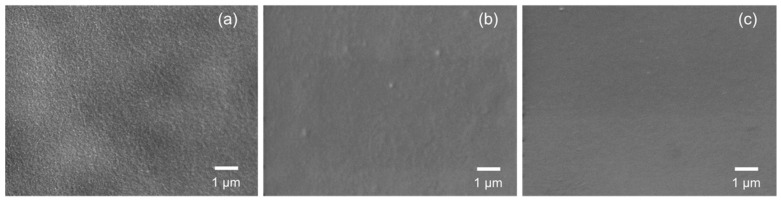
SEM images of the top surface view of an (**a**) as-synthesized mesoporous γ-Al_2_O_3_ layer and 500 °C-synthesized membrane further modified with PCS (**b**) before and (**c**) after gas permeation measurements under p/p_0_(H_2_O) up to 1 at 50 °C.

**Table 1 membranes-10-00258-t001:** Activation energies for He and H_2_ permeations through the supported mesoporous γ-Al_2_O_3_ membrane after modification with AHPCS and subsequent heat treatment at 300–500 °C.

SiCH Hybrid/γ-Al_2_O_3_ Composite Membrane Synthesis Temp.	Activation Energy (*E_a_*)/kJ mol^−1^	
	He	H_2_
300 °C	20.3	17.0
400 °C	9.8	8.6
500 °C	1.4	0.5

**Table 2 membranes-10-00258-t002:** Gas permeation data of 500 °C-synthesized membrane further modified with PCS measured at 50 °C under p/p_0_ (H_2_O) =1.0 using a H_2_-N_2_ (2:1) mixed feed gas.

Feed Side		Permeate Side				Separation Factor (SF)
Flow Rate, *V*_F_/× 10^−5^ mol s^−1^		Flow Rate, *V*/× 10^−5^ mol s^−1^		Flux, *J*/× 10^−5^ mol m^−2^ s^−1^		
*V* _F, H2_	*V* _F, N2_	*V* _H2_	*V* _N2_	*J* _H2_	*J* _N2_	
14.7	7.4	2.2	0.043	2971.8	56.6	26

**Table 3 membranes-10-00258-t003:** H_2_-permselectivities of composite membrane in this study and those of other membranes evaluated at *T* ≤ 50 °C under the dry condition.

No.	Membranes	Temp./°C	H_2_ Permeance	H_2_/X Selectivity		Ref.
			/mol m^−2^ s^−1^ Pa^−1^	α(H_2_/X)	X	
#01	500 °C modified with PCS, Dry	50	8.4 × 10^−7^	29.6	N_2_	This study
	Wet, p/p_0_(H_2_O) = 1.0	50	3.5 × 10^−7^	36.1	N_2_	
#02	SiO_2_-based organic–inorganic hybrid (BTESE)	40	7.66 × 10^−7^	−20	N_2_	[68]
#03	SiO_2_-based organic–inorganic hybrid (BTESM)	50	1.79 × 10^−6^	12	N_2_	
#04	zeolite-β/CMS	30	2.64 × 10^−9^	97	N_2_	[69,70]
		30	1.95 × 10^−8^	67.3	N_2_	
#05	zeolite-Y/CMS	30	2.77 × 10^−9^	100.2	N_2_	
		30	1.96 × 10^−8^	61	N_2_	
#06	PFDA-based liquid membrane	30	2.4 × 10^−9^	10	O_2_	[18]
#07	graphene nanosheet	RT	5.9 × 10^−7^	16.5	N_2_	[71,72]
#08	ZIF nanosheet membrane	30	2.04 × 10^−7^	66.55	N_2_	[73,74]
#09	ultrathin graphene oxide	20	−1 × 10^−7^	−900	N_2_	[75,76]
#10	microporous polymer (PIM-EA(H_2_)-TB)	30	(1.7 × 10^−12^) *	22	N_2_	[77]
#11	microporous polymer (PIM-Trip(Me_2_)-TB)	25	(1.8 × 10^−12^) *	21.4	N_2_	[78]

*: H_2_ permeability/mol m^−1^ s^−1^ Pa^−1^.

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
