# Peer review of "Hydrogen Selective SiCH Inorganic–Organic Hybrid/γ-Al2O3 Composite Membranes"

_membranes, 2020, doi:10.3390/membranes10100258_

Round 1

Reviewer 1 Report

In this study, a supported mesoporous γ-Al2O3 membrane was modified with allyl-hydrid/ polycarbosilane as a preceramic polymer and subsequently heat-treated in Ar to deliver a ternary SiCH organic-inorganic hybrid/γ-Al2O3 composite membrane. Relations between the polymer/hybrid conversion temperature, hydrophobicity and H2 affinity of the polymer-derived SiCH hybrids were studied to functionalize the composite membranes as H2-selective under saturated water vapor partial pressure at 50 °C. There are several issues that needs to be fixed before the acceptance of this paper, particularly the stability/reusability of the developed membrane as well as long term performance of the membrane. I believe that the authors have to re-do characterization and stability experiments. In the introduction section, please identify the research gaps and how the proposed membrane would address these issues. This could be added as the second last paragraph of the introduction section. The M&M section is well organised. Please add information regarding the purity of the chemicals used in this study. In addition, please clarify if the experiments were replicated to ensure reproducibility. Please explain briefly – the basis of selecting a certain molar ratio of gases in the feed. Is it environmentally relevant and what is the area/industry of application for the developed membrane? A range of results are shown here in this study, which are interesting. However, please provide a table showing the comparison of the membrane performance in this study with that in previous studies. The characterization component of this MS is weak. Ideally, there should be information on the surface roughness and surface charge of the membrane before and after the experiments. This will help to understand to long term performance/stability of the membrane. Why ceramic membrane was selected in this study and what would be the performance without its modification. Please check if the authors really want to use the term ‘hydrid’ in the abstract and elsewhere in the manuscript.

Author Response

Response to the Comments from Reviewer 1#

Thank you for the valuable comments. We have considered all of your comments and revised our manuscript with respect to your suggestion. Your valuable suggestions and comments were carefully considered during the manuscript revision. All changes performed in the text are highlighted using blue word.

Please find an attached file, our reply to your valuable suggestions and comments are summarized in the file.

Sincerely yours,

Yuji IWAMOTO

On behalf of all co-authors

Nagoya, Japan, September 21st, 2020.

Reviewer 2 Report

In this manuscript, the authors synthesized a series of SiCH organic-inorganic hybrid membranes. Their chemical, thermal, and gas permeance properties are studied. The overall flow of the manuscript is good and a minor revision is needed before publication.

  1. In the first paragraph of the Introduction, the authors may introduce other H2 production approaches, and compare the pros and cons of each method.
  2. In the second paragraph, the authors may also compare the pros and cons of membrane separation technique versus other techniques.
  3. Contact angle tests may be a good addition to hydrophobicity study.
  4. How is the long-term stability and reproducibility of the membranes? The authors may provide long-term and repeated tests of gas permeance of the samples.

Author Response

Response to the Comments from Reviewer 2#

Thank you for the valuable comments. We have considered all of your comments and revised our manuscript with respect to your suggestion. Your valuable suggestions and comments were carefully considered during the manuscript revision. All changes performed in the text are highlighted using blue word.

Please find an attached file.

Our reply to your valuable suggestions and comments are summarized and shown in the attached file.

Sincerely yours,

Yuji IWAMOTO

On behalf of all co-authors

Nagoya, Japan, September 21st, 2020.

Reviewer 3 Report

This problem is relevant for journal scope. The concept and aim are clearly defined. The manuscript is well written and really informative. The manuscript follows the formal regulations of journal.

I suggest the acceptance after major revision.

  1. Please cite more papers from this journal at the last two years in the similar topic of this research.
  2. Please add Nomenclature part to the manuscript.
  3. The main contribution and novelty of the work are not identified. The scientific justification for this work should be added.
  4. What do you think about the chance for industrial application of your offered process?
  5. Please compare your results with literature results.
  6. Please mention some information about separation efficiency of membrane. Please add the separation factor and if it available (partial) flux values too.

Author Response

Response to the Comments from Reviewer 3#

Thank you for the valuable comments. We have considered all of your comments and revised our manuscript with respect to your suggestion. Your valuable suggestions and comments were carefully considered during the manuscript revision. All changes performed in the text are highlighted using blue word.

Please find an attached file.

Our reply to your valuable suggestions and comments are summarized and shown in the attached file.

Sincerely yours,

Yuji IWAMOTO

On behalf of all co-authors

Nagoya, Japan, September 21st, 2020.

Round 2

Reviewer 1 Report

The manuscript may be accepted for publication. 

Reviewer 3 Report

I have studied the manuscript and the answers. In my opinion the answers are professionally well-founded. I suggest the acceptance in this present form for publication.